# Accounting for Uncertainty in Clonal Phylogeny Reconstruction

**Maitena Tellaetxe-Abete, Borja Calvo**
University of the Basque Country
Donostia/San Sebastian, 20008
{maitena.tellaetxe,borja.calvo}@ehu.eus

## Abstract

Intratumoral heterogeneity in cancer arises from the evolutionary accumulation of genetic mutations, leading to multiple clones within a single tumor. The Clonal Deconvolution and Evolution Problem addresses the reconstruction of these distinct clonal subpopulations and their ancestral relationships using mutation frequency estimates with varying levels of reliability. In this project, we propose an Iterated Local Search algorithm with two objective functions: one that treats all information uniformly and another that accounts for the uncertainty associated with each instance element by giving greater weight to more reliable positions. Our ultimate goal is to determine the conditions under which leveraging uncertainty enhances the performance of metaheuristic algorithms.

## 1 Motivation

Cancer develops through an evolutionary mechanism, accumulating genetic mutations over successive cell divisions [8]. When a cell acquires a new mutation and divides, it creates a subpopulation—called a clone—whose cells all share that mutation. As additional mutations emerge, multiple distinct clones appear, each with a unique set of mutations [10]. This intratumoral heterogeneity has profound implications not only for our understanding of the disease but also for effective cancer management, as different clones within a tumor may respond differently to treatments, resulting in therapeutic resistance and disease relapse [9].

Analyzing the tumor phylogeny—the evolutionary history of cancer cells—has become a valuable approach to understanding how mutations accumulate and propagate within a tumor. From a computational perspective, a tumor phylogeny, or, more specifically, a clonal tree, can be viewed as a rooted tree in which each node represents a clone carrying a particular combination of mutations, and the edges denote ancestral relationships. The root of this tree typically corresponds to the clone that first initiated the tumor.

However, detecting and distinguishing individual clones is challenging because, in most cases, cancer genomics studies rely on bulk DNA sequencing data [4, 11]. In a bulk sequencing approach, one or more tumor biopsies are collected, and the DNA from billions of cells within these samples is sequenced simultaneously to identify the mutations present [12]. As a result, each mutation is assigned a Variant Allele Frequency (VAF) value, which represents the ratio of mutant reads to total reads and serves as an estimate of the fraction of cells in the sample carrying that mutation. Although bulk sequencing is an essential tool in cancer research, it complicates phylogeny reconstruction because each biopsy typically contains an unknown mixture of multiple clones. Moreover, mutations can be shared across different clones, so the VAF value for any given mutation reflects the cumulative signal of all clones that harbor it. Consequently, it is not straightforward to infer a clonal tree directly from bulk sequencing data without first determining the contribution of each clone to the observed VAF values.

XVI XVI Congreso Español de Metaheurísticas, Algoritmos Evolutivos y Bioinspirados (maeb 2025).

Within this context emerges the Clonal Deconvolution and Evolution Problem (CDEP), which is inherently an optimization problem. Here, each instance is represented by a matrix of VAF values, where each value estimates the mutation fraction in a tumor sample. The goal is to determine the tumor's clonal structure—including the number, proportion, and mutational composition of the clones in each sample—and reconstruct the clonal phylogeny that explains how these distinct populations evolved.

It is important to note, however, that the DNA sequencing process from which these VAF values are derived introduces noise, causing the values to deviate from the true mutation fractions. Consequently, even if a method provides a mathematically optimal solution, it may not translate into an optimal solution in practice. Furthermore, the extent of this noise can vary across different regions of the genome, rendering some VAF values more reliable than others. This means that, even within a single instance, the reliability of the VAF values may vary, adding an extra layer of complexity to the reconstruction process.

## 2    Working Hypothesis

The fact that not all VAF values are equally reliable suggests that solving the CDEP for a given instance while treating every element as equally informative, rather than prioritizing more reliable observations, may lead to suboptimal decisions and, consequently, inaccurate clonal reconstructions. This consideration motivates us to investigate the impact of accounting for varying levels of uncertainty (or, equivalently, reliability[1]) within an instance when guiding metaheuristic algorithms to solve the CDEP.

In this context, we formulate the following hypothesis:

> *When the uncertainty is not uniformly distributed within an instance, an objective function that accounts for these differences in the reliability of the VAF estimator can guide metaheuristic algorithms for the CDEP more efficiently.*

## 3    Objectives

Based on the hypothesis presented in the previous section, we propose the following objectives:

**Objective 1.** Determine whether explicitly accounting for uncertainty, i.e., varying reliability of VAF measurements within an instance, can indeed improve the quality of clonal reconstructions obtained by metaheuristic algorithms.

**Objective 2.** If accounting for uncertainty proves beneficial, identify the specific conditions or instance features under which these considerations lead to more accurate solutions.

## 4    Methodology

To test our hypothesis, we have designed an observational study to investigate whether and how accounting for VAF uncertainty can lead to more accurate clonal reconstructions, particularly when using metaheuristic algorithms. This study is divided into two main phases. The first phase involves generating various synthetic datasets with different levels and distribution of uncertainty in the VAF values and then solving the CDEP for each instance using both a classical algorithm and an algorithm that explicitly accounts for uncertainty. The second phase comprises a meta-analysis in which we characterize the instances and conduct a machine learning–based analysis to determine the conditions under which accounting for uncertainty provides an advantage. The following subsections describe these phases in greater detail.

### 4.1    First Stage: Instance Generation and Solving the CDEP

In this first stage, we introduce a standard Iterated Local Search (ILS) algorithm as a baseline to explore the potential benefits of leveraging reliability. It is important to note that our goal is not to

---

[1]Both uncertainty and reliability refer, in this context, to the amount of empirical evidence influencing the accuracy of the VAF estimator. Throughout this work, we use both terms interchangeably to refer to this concept.

develop a novel algorithm but rather use it as a tool to analyze the potential of leveraging uncertainty, which we evaluate through an objective function. Since the focus of this study is on analyzing the role of uncertainty rather than on developing advanced methods for solving the CDEP, a basic metaheuristic is sufficient for this purpose. This algorithm is applied with two objective functions: one that considers only the VAF values, and a second function that also incorporates the level of uncertainty associated with each VAF value. We then describe a methodology for generating instances that vary in both the amount and distribution of uncertainty, followed by the experimental protocol used to compare the performance of the two objective functions across these instances.

### 4.1.1 An Iterated Local Search Algorithm

The mathematical foundation of the CDEP on which we build our approach is an existing formulation known as the Variant Allele Frequency Factorization Problem (VAFFP) [3].

Given $m$ tumor samples and $n$ mutations identified across these samples, we define a matrix $\boldsymbol{F} \in [0,1]^{m \times n}$, where each element $f_{ij}$ represents the VAF value for mutation $j$ in sample $i$. The VAFFP seeks to decompose this input matrix $\boldsymbol{F}$ into two matrices, $\boldsymbol{U} \in [0,1]^{m \times n}$ and $\boldsymbol{B} \in \{0,1\}^{n \times n}$, such that $\boldsymbol{F} = \boldsymbol{U} \cdot \boldsymbol{B}$. Here, $\boldsymbol{B}$ represents the clonal phylogeny, where $b_{ij} = 1$ iff clone $i$ contains mutation $j$ [5], while $\boldsymbol{U}$ captures the proportion of each clone in each tumor sample.

We address this factorization by searching over possible phylogenies represented by a $\boldsymbol{B}$ matrix using an Iterated Local Search (ILS) algorithm. This algorithm starts with a semi-random, GRASP-inspired phylogeny and iteratively refines the solution. At each iteration, it explores the neighborhood of the current solution through a Subtree Prune and Regraft (SPR) move, which involves cutting one edge in the tree and reattaching the resulting subtree to another node in the remaining tree [2, 1]. The algorithm then moves to the first neighboring solution encountered that improves the objective function. If no improvement is found, a perturbation step applies several random SPR operations to escape from local optima. This process continues until a global lower bound for the objective function is reached or a computational budget is exceeded.

To evaluate each candidate solution, we propose two different objective functions.

**Standard Objective Function**  The standard objective function, a classical approach in the literature, treats all VAF values—estimates of the mutation fraction—of an instance equally, ignoring the differences in certainty among them.

Given a clonal phylogeny represented by $\boldsymbol{B}$, we first calculate the uniquely defined matrix $\boldsymbol{U}' = \boldsymbol{F} \cdot \boldsymbol{B}^{-1}$. This step is straightforward as $\boldsymbol{B}$ is always an invertible matrix [3]. We then obtain $\boldsymbol{U}$ by coercing any negative entries in $\boldsymbol{U}'$ to 0 and normalizing each row to sum to 1, and we compute $\tilde{\boldsymbol{F}} = \boldsymbol{U} \cdot \boldsymbol{B}$. Finally, the objective function value is calculated as the mean absolute error between the $\boldsymbol{F}$ matrix that represents the instance and the $\tilde{\boldsymbol{F}}$ matrix:

$$g_S(\boldsymbol{B}) = \frac{1}{m \cdot n} \sum_{i=1}^{m} \sum_{j=1}^{n} \left| f_{ij} - \tilde{f}_{ij} \right|$$

**Uncertainty-Aware Objective Function**  The second objective function is based on a Bayesian approximation that incorporates uncertainty in the VAF values.

In general, confidence in a VAF value depends on the total number of observations or reads at that genomic position: more reads mean the observed frequency is likely closer to the true mutation rate (recall that the VAF is the ratio of mutant reads to total reads). Furthermore, when uncertainty is high, a position's contribution to the objective function should be minimal; when uncertainty is low, its impact should be larger; and as uncertainty in the instance decreases, the objective function should converge to the error.

The initial evaluation steps for this function are identical to those for the standard function, generating $\tilde{\boldsymbol{F}}$ from $\boldsymbol{B}$ as described in Section 4.1.1. Now, each genomic position has $r_{ij}$ total reads and $r_{ij}^a$ reads supporting the mutation. We assume that these observations follow a binomial distribution, $r_{ij}^a \sim Binomial(r_{ij}, \varphi_{ij})$, where $\varphi_{ij}$ represents the unknown probability of observing the mutation

at that position, which is the value we seek to estimate. We place a uniform $Beta(1,1)$ prior on $\varphi_{ij}$, yielding a posterior distribution given by $P(\varphi_{ij} \mid r_{ij}, r_{ij}^a) \sim Beta\left(r_{ij}^a + 1, r_{ij} - r_{ij}^a + 1\right)$.

Under this model, we evaluate each element $\tilde{f}_{ij}$ using the expected value of the absolute difference between $\varphi_{ij}$ and $\tilde{f}_{ij}$ weighted by a correction factor to account for the variance of this difference, as follows:

$$h(\tilde{f}_{ij}) = \frac{1}{a} \cdot E(\epsilon_{ij})\ (a - VAR(\epsilon_{ij})), \quad \text{where} \quad \epsilon_{ij} = \left|\varphi_{ij} - \tilde{f}_{ij}\right|$$

Here, $a$ denotes the variance of the function $\epsilon_{ij}$ in the least informative case, i.e., when $r_{ij} = 0$, which is the scenario where we obtain the maximum variance for $\epsilon_{ij}$. In such a situation, any value of $\varphi_{ij}$ is equally probable, meaning the position is non-informative. With this formula, we ensure that $h(\tilde{f}_{ij})$ takes on the value of 0 in that case. As $r_{ij}$ increases, the variance of $\varphi_{ij}$ (and thus, of $\epsilon_{ij}$) decreases and approaches 0, causing the function to converge towards the expected value of the difference, and consequently, to the value provided by the standard function.

Finally, the objective function value is given by mean value of all the $h(\tilde{f}_{ij})$ values:

$$g_U\left(\boldsymbol{B}\right) = \frac{1}{m \cdot n} \sum_{i=1}^{m} \sum_{j=1}^{n} h(\tilde{f}_{ij})$$

### 4.1.2 Instance Generation

Our procedure for generating synthetic instances to evaluate our hypothesis begins with the simulation of clonal trees with varied attributes—such as the number of clones, topology, and sample size—and the generation of corresponding total and mutation read count matrices, which yield a noise-free VAF matrix $\boldsymbol{F}$ for each instance. Our approach distinguishes between positions with high uncertainty (10 total reads) and low uncertainty (1,000 total reads). We then introduce errors that simulate typical sequencing inaccuracies, which have a greater detrimental impact on VAF values at high-uncertainty positions than at low-uncertainty ones. By systematically varying both the quantity (10%, 20%, 50%, and 75%) and the distribution—across the entire matrix, as well as within individual rows (samples) and columns (mutations)—of these high-uncertainty positions, we aim to create scenarios where leveraging uncertainty may offer advantages, have no discernible benefits, or even lead to worse outcomes. This process results in approximately 3,000 noisy instances, ensuring a broad and diverse range of conditions to test our hypothesis.

### 4.1.3 Experimental Protocol

Our experimental protocol involves running the ILS algorithm five times on each instance for each objective function. Based on these runs, we determine which method produces the best clonal reconstruction, or whether their performance is equivalent, as defined by a region of practical equivalence (ROPE) [6].

### 4.2 Second Stage: Identification of Contexts Where Accounting for Uncertainty Yields Benefits

To analyze the conditions under which each objective function performs best, we record a set of descriptors that capture various aspects of the instances. These include instance size; statistics derived from VAF values (such as the prevalence of ties, zeros, and the overall average); and read count metrics computed both overall and by samples and mutations. Furthermore, we quantify how uncertainty is distributed within the phylogeny. For example, we assess whether high-read positions tend to be associated with higher or lower VAF values, whether they correspond to nodes with many (or few) children or are uniformly distributed in that regard, and how node position—whether near the top or bottom of the phylogeny—relates to its read count. We then use these descriptors to train a supervised classification model that predicts the best-performing objective function for each instance. Finally, we employ the SHapley Additive exPlanations (SHAP) method [7] to examine how the model makes its predictions, shedding light on the instance characteristics that lead one objective function to outperform the other.

## Acknowledgments and Disclosure of Funding

This work was supported by Departamento de Educación, Universidades e Investigación of the Basque Government [PRE_2020_2_0101]; Ministerio de Economía, Industria y Competitividad (MINECO) of the Spanish Central Government [PID2022-137442NB-I00], and Departamento de Industria of the Basque Government [IT1504-22 and ELKARTEK Programme].

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
