# OpenReview forum: "Accounting for Uncertainty in Clonal Phylogeny Reconstruction"
_MAEB/2025/Projects_Track — MAEB 2025 Proyectos_

### Official Review · Reviewer_bFC1 · 2025-03-17
**Review for Accounting for Uncertainty in Clonal Phylogeny Reconstruction**

**Rating:** 5
**Confidence:** 4

**Review:**

The work proposes to account for uncertainty in the fitness functions of MHs when dealing with the clonal phylogeny reconstruction problem.

The problem is well stated, as are the hypothesis and objectives. The methodology, which involves developing experiments with a metaheuristic under two scenarios—one with a standard fitness function and another with one that accounts for uncertainty—seems appropriate. Nevertheless, some questions arise:

The work states, "a (single) basic metaheuristic is sufficient for this purpose." I am not sure about this. Could not, perhaps, a single bad metaheuristic provide misleading results in this study? By bad, I mean a metaheuristic that is not sufficiently appropriate. For instance, if your metaheuristic were unable to provide acceptable clonal reconstructions in most cases, you might be unable to detect performance differences between using one fitness function or the other.

The work is based on synthetic instances. I assume this is because the real clonal phylogeny is not known for actual instances. Is that really the case? Otherwise, including real data would clearly enrich the study.

I have not understood why what appears to be uniform noise (Section 4.1.2) could be used to detect non-uniform differences (Section 4.2).

Finally, it seems to me that the project is, somehow, not very ambitious. In particular, accounting for uncertainty when addressing a problem that inherently involves uncertainty levels appears likely to yield a nearly trivial conclusion: "It is smart to adapt to the problem's characteristics." Nevertheless, I must admit that the proposed research path is indeed the right approach: "People have not accounted for uncertainty in this problem. We are the first ones."

---

### Official Review · Reviewer_eNVD · 2025-03-17
**Accounting for Uncertainty in Clonal Phylogeny Reconstruction**

**Rating:** 3
**Confidence:** 1

**Review:**

This is a project paper aiming to investigate a bi-objective local search algorithm  to consider in Uncertainty in Clonal Phylogeny
Reconstruction. The paper is well-written. I am personally not expert in this field, and, while I found that the aspects of the paper make sense, I cannot judge with confidence the paper.

---

### Official Review · Reviewer_KTHn · 2025-03-19
**Review of Project "Accounting for Uncertainty in Clonal Phylogeny Reconstruction"**

**Rating:** 5
**Confidence:** 4

**Review:**

In my opinion, the project is very interesting and deserves to be developed and evaluated. I would like to encourage the authors to consider, if possible, some minor comments:
1) The number of repetitions of the experiments should be higher than 5 (advisable, from a statistical perspective, 30 times), given that randomness play a crucial role in aspects of the optimization process.
2) You should give more details about the initialization process ("a semi-random, GRASP-inspired phylogeny").
3) About the second stage, authors should include some validation and test procedure, to ensure that the machine learning model is not overfitting the data.

---

### Decision · Program_Chairs · 2025-03-19

Accept